

# Long-term trends in the PM₂.₅- and O₃-related mortality burdens in the United States under emission reductions from 1990 to 2010

Yuqiang Zhang[1,*], J. Jason West[2], Rohit Mathur[3], Jia Xing[4], Christian Hogrefe[3], Shawn J. Roselle[3], Jesse O. Bash[3], Jonathan E. Pleim[3], Chuen-Meei Gan[5], David C. Wong[3]

[1]Oak Ridge Institute for Science and Education (ORISE) Fellowship Participant at US Environmental Protection Agency, Research Triangle Park, NC 27711, USA

[2]Department of Environmental Sciences and Engineering, University of North Carolina at Chapel Hill, Chapel Hill, NC 27599, USA

[3]Computational Exposure Division, National Exposure Research Laboratory, Office of Research and Development, US Environmental Protection Agency, Research Triangle Park, NC 27711, USA

[4]State Key Joint Laboratory of Environmental Simulation and Pollution Control, School of Environment, Tsinghua University, Beijing 100084, China

[5]CSC Government Solutions LLC, A CSRA Company, Research Triangle Park, NC, 27709, USA

[*]now at: Nicholas School of the Environment, Duke University, Durham, NC 27710, USA

*Correspondence to*: Yuqiang Zhang (yuqiangzhang.thu@gmail.com; yuqiang.zhang@duke.edu)

**Abstract.** Concentrations of both fine particulate matter ($PM_{2.5}$) and ozone ($O_3$) in the United States (US) have decreased significantly since 1990, mainly because of air quality regulations. These air pollutants are associated with premature death. Here we quantify the annual mortality burdens from $PM_{2.5}$ and $O_3$ in the US from 1990 to 2010, estimate trends and inter-annual variability, and evaluate the contributions to those trends from changes in pollutant concentrations, population, and baseline mortality rates. We use a fine-resolution (36 km) self-consistent 21-year simulation of air pollutant concentrations in the US from 1990 to 2010, a health impact function, and annual county-level population and baseline mortality rate estimates. From 1990 to 2010, the modeled population-weighted annual $PM_{2.5}$ decreased by 39%, and summertime (April to September) 1hr average daily maximum $O_3$ decreased by 9% from 1990 to 2010. The $PM_{2.5}$-related mortality burden from ischemic heart disease, chronic obstructive pulmonary disease, lung cancer, and stroke, steadily decreased by 53% from 123,700 deaths $yr^{-1}$ (95% confidence interval, 70,800-178,100) in 1990 to 58,600 deaths $yr^{-1}$ (24,900-98,500) in 2010. The $PM_{2.5}$-related mortality burden would have decreased by only 24% from 1990 to 2010 if the $PM_{2.5}$ concentrations had stayed at the 1990 level, due to decreases in baseline mortality rates for major diseases affected by $PM_{2.5}$. The mortality burden associated with $O_3$ from chronic respiratory disease increased by 13% from 10,900 deaths $yr^{-1}$ (3,700-17,500) in 1990 to 12,300 deaths $yr^{-1}$ (4,100-19,800) in 2010, mainly caused by increases in the baseline mortality rates and population, despite decreases in $O_3$ concentration. The $O_3$-related mortality burden would have increased by 55% from 1990 to 2010 if the $O_3$ concentrations had stayed at the 1990 level. The detrended annual $O_3$ mortality burden has larger inter-annual variability (coefficient of variation



of 12%) than the PM$_{2.5}$-related burden (4%), mainly from the inter-annual variation of O$_3$ concentration. We conclude that air quality improvements have significantly decreased the mortality burden, avoiding roughly 35,800 (38%) PM$_{2.5}$-related deaths and 4,600 (27%) O$_3$-related deaths in 2010, compared to the case if air quality had stayed at 1990 levels.

## 1 Introduction

The 2015 Global Burden of Disease (GBD) study lists air pollution as the fourth-highest ranking global mortality risk factor (GBD 2016). The most recent GBD estimates that exposure to ambient particulate matter (particulate size less than 2.5 µm, PM$_{2.5}$) causes 4.2 million (95% confidence interval (CI), 3.7-4.8 million) deaths globally, with an additional 254,000 (97,000-422,000) deaths globally caused by exposure to ozone (O$_3$) (Cohen et al., 2017). For the United States (US), the same study estimated a mortality burden attributable to ambient PM$_{2.5}$ of 88,400 (66,800-115,000) deaths, and 11,700

(4,400–19,600) deaths attributable to O$_3$ in 2015 (Cohen et al., 2017). Adopting similar methods, the US burden of disease study lists ambient PM$_{2.5}$ and O$_3$ pollution as the 8[th] and 15[th] leading risk factors in the US in 2010 (Murry et al., 2013). Over recent decades, emissions of air pollutants within the US have significantly decreased and air quality has improved. For example, between 1990 and 2010, total US anthropogenic emissions are estimated to have declined by 48%, 49%, 67%, 60% and 34%, for NO$_x$ (=NO + NO$_2$), non-methane volatile organic compounds (NMVOC), SO$_2$, CO and PM$_{2.5}$ respectively

(Xing et al., 2013). EPA observations sites showed that the US average annual PM$_{2.5}$ concentration declined by 42% from 2000 to 2016, and maximum daily 8-h average (MDA8) O$_3$ declined by 22% from 1990 to 2016 (US EPA, 2017). These air quality improvements were mainly driven by ambient air quality standards, and federal and state implementation of stationary and mobile source regulations, especially the 1990 Clean Air Act (CAA) Amendments, the 2002 NO$_x$ State Implementation Plans (SIP) Call, and the Cross-State Air Pollution Rule (US EPA, 2011). Other factors that improved air

quality include technological development that made emission control technologies more effective and less expensive, and changes in the energy system such as the recent growth of natural gas and wind that has displaced coal for electricity generation. These decreased concentrations are expected to have brought substantial benefits for public health in the US, but assessing the health benefits requires quantification of changes in human exposure relating where air quality has improved with where people live.

Several recent studies have assessed the global (GBD 2015, 2016; Lelieveld et al., 2015; Silva et al., 2013, 2016b) or national (Fann et al., 2012; Punger and West, 2013) burdens of disease attributable to air pollution. However, less effort has been made to understand how these burdens evolve over time. Cohen et al. (2017) used estimates of air pollutant concentrations from a combination of air quality model simulations, satellite and surface observations to study the global and national temporal trends of the burdens of disease attributable to ambient PM$_{2.5}$ and O$_3$ (only model results were used for O$_3$),

at 5-yr intervals from 1990 to 2015. Two other studies (Butt et al., 2017; Wang et al., 2017) used coarse resolution model simulations to study the global/ hemispheric PM$_{2.5}$ mortality burdens for the past few decades. Fann et al. (2017) estimated the annual mean PM$_{2.5}$ concentration in the US from monitoring data and its all-cause mortality burden in 1980, 1990, 2000



and 2010. Epidemiological studies have also inferred how health effects have changed through time in the US (Correia, et al., 2013; Pope et al., 2009). However, previous studies have not estimated mortality burdens for both $PM_{2.5}$ and $O_3$ in the US for several years in succession, investigated the different drivers for mortality trends, or the inter-annual variability of the mortality burdens. In the US, such changes in the air pollution mortality burdens can support decision making on air

pollution control policies. For the public, analyzing trends can effectively illustrate the benefits of past air pollution controls, as well as the challenges for future policy efforts.

Here we aim to quantify air pollution-related mortality in the continental US in each year from 1990 to 2010, mainly to understand the trends over this time period. We also analyze the contributions of changes in air pollutant concentrations, population, and baseline mortality rates to the overall trend, and analyze the inter-annual variability in mortality burdens

estimates. To achieve this, we use a 21-year (1990-2010) model simulation of $PM_{2.5}$ and $O_3$ concentrations over the Continental US (CONUS) from 1990 to 2010. We also use annual county-level baseline mortality rates and population archived by the US Centers for Disease Control (CDC WONDER, https://wonder.cdc.gov/mortSQL.html).

## 2 Methodology

### 2.1 Air quality simulations

We use simulations of air quality over the CONUS from 1990 to 2010 by Gan et al. (2015, 2016). These simulations used the WRF-CMAQ model which coupled the Weather Research and Forecasting (WRF, v3.4) model and the Community Multiscale Air Quality (CMAQ, v5.02) model (Wong et al., 2012), covering the CONUS at 36km×36km. These simulations used an internally-consistent emission inventory from 1990 to 2010 for the US developed by Xing et al. (2013), three-dimensional meteorological fields constrained by reanalysis of available surface and aloft measurements of the atmospheric state, and time-

varying lateral boundary conditions provided by the hemispheric CMAQ (Mathur et al., 2017; Xing et al., 2015). The model system used the CB05 chemical mechanism with the AERO6 module for aerosols.

Gan et al. (2015, 2016) showed that the modeled trends of total $PM_{2.5}$ and its components generally matched both the CASTNET (Clean Air Status and Trend Network) and IMPROVE (Interagency Monitoring of Protection Visual) observational data from 1995 to 2010, with correlation coefficients usually larger than 0.87 for total $PM_{2.5}$ and its components. Moreover,

the trends for $PM_{2.5}$ and its species were similar in direction and magnitude (Gan et al., 2015). There was a small or nearly no trend for $PM_{2.5}$ in western US for both the model and observations, while a dramatic decreasing trend in the eastern US, with a larger decreasing trend from the model (-0.44 µg m$^{-3}$ yr$^{-1}$) than from the IMPROVE observations (-0.30 µg m$^{-3}$ yr$^{-1}$).

For $O_3$, Astitha et al. (2017) used dynamical evaluation methods, and showed that the simulated $O_3$ trends generally agreed very well with the observed downward trends, especially for the period from 2000 to 2010, albeit underestimating trends over

some regions, for both the May to September average of daily maximum 8-hr (MDA8) and annual 4[th] highest $O_3$. From 2000 to 2010, the regional trends for the 4[th] highest $O_3$ from the model (observations) were -0.80 ppbv yr$^{-1}$ (-0.73 ppbv yr$^{-1}$) for the



Southwest, -1.14 (-1.53) for Southcentral, -1.31 (-1.66) for Southeast, -1.46 (-1.61) for Midwest, -1.35 (-1.79) for Northeast, and -1.11 (-1.40) for CONUS.

**2.2 Mortality burden attributable to ambient air pollution**

The mortality burdens attributable to ambient $PM_{2.5}$ and $O_3$ ($\Delta Mort$) are estimated using the health impact function (HIF) following Eq. (1):

$$\Delta Mort = y_0 \times AF \times Pop \tag{1}$$

where $y_0$ is the baseline mortality rate for specific diseases, $AF$ is the attributable fraction calculated as $1 - 1/RR$, with $RR$ as the relative risk of death from a specific disease, and $Pop$ is exposed population age 25 years and greater.

For $PM_{2.5}$, $RR$ is calculated using the integrated exposure–response (IER) model (Burnett et al., 2014), which has been extensively used by recent studies, including Liu et al. (2017), Silva et al. (2016a, b), Wang et al. (2017), and World Health Organization (2016). The $RR$ is calculated as a function of $PM_{2.5}$ concentration following Eq. (2):

$$for \ C < C_0, RR(C) = 1$$
$$for \ C \geq C_0, RR(C) = 1 + \alpha \times (1 - \exp(-\gamma \times (C - C_0)^\delta)) \tag{2}$$

where $C$ is the annual average ambient $PM_{2.5}$ concentration, $C_0$ is the $PM_{2.5}$ threshold concentration (5.8-8.0 µg/m³), below which no additional risk is assumed, and the parameter values of $\alpha$, $\gamma$, and $\delta$ are given by distributions (Burnett et al., 2014). For this study, the RRs are downloaded from GBD website (Global Health Data Exchange (GHDx) 2013).

For the $O_3$-related mortality burden, RR$= exp^{\beta\Delta X}$ , where $\beta$ is the concentration response factor, and $\Delta X$ is the difference in $O_3$ concentration (summertime 1hr daily maximum $O_3$) between the current year (1990 to 2010) and the low-concentration threshold. For $RR$, we use the value of 1.040 (with 95% CI: 1.013, 1.067) from Jerrett et al. (2009) following recent studies (e.g, Cohen et al., 2017; GBD 2015, 2016; Lim et al., 2012). Turner et al. (2016) found a larger $RR$ for respiratory mortality ($RR$, 1.12; 95% CI, 1.08-1.16) associated with the annual average of MDA8 $O_3$, and using these results would likely lead to a larger $O_3$ mortality burden (Malley et al., 2017). We account for all chronic respiratory disease (RESP), to be consistent with Jerrett et al. (2009). The low-concentration threshold is the average of 33.3-41.9 ppbv $O_3$ indicated by Lim et al., (2012), that is, 37.6 ppbv (Lelieveld et al., 2015). We use adults above 25 years, to be comparable with other calculations of $PM_{2.5}$ mortality burden following Silva et al. (2016a,b), even though the estimated RR from Jerrett et al. (2009) were for adults above 30 old only. Uncertainties in air pollution-related mortality burden calculations are based on the uncertainty in RRs only, ignoring those in modeled air pollutant concentrations, and population and baseline mortality rates.

To estimate the annual baseline mortality rates ($y_0$) for each disease associated with $PM_{2.5}$ (chronic obstructive pulmonary disease, COPD; ischemic heart disease, IHD; lung cancer, LC; cerebrovascular disease and ischemic stroke, STROKE) and $O_3$ (chronic respiratory disease, RESP), we acquire US county-level specific causes of mortality data for each year from the National Center for Health Statistics (NCHS) (CDC, 2017). We then regrid the county-level mortality data to each model grid cell at 36km×36km. The specific causes of mortality data for some counties are sometimes suppressed when the total deaths



are lower than 10 per year to protect privacy (Jian et al., 2016), missing or considered as "unreliable" when the total deaths are less than 20 per year, and are corrected following established procedures (BenMAP, 2017; Fann et al., 2017; also see Supporting Information).

Definitions of each disease follow the Global Burden of Disease (GBD) study (Lim et al., 2012; supporting Table S1). Note
that the CDC changed the disease codes from the International Classification of Diseases 9th Revision (ICD 9) in 1998 to ICD10 in 1999, and there were discontinuities in the death counts of specific diseases (Anderson et al., 2001; Anderson & Rosenberg, 2003). To account for the discontinuities, we group the total deaths for each of the five diseases for ICD9 and ICD10 using the results of Anderson et al., (2001) and Anderson & Rosenberg (2003) who reported deaths for 135 specific causes in 1996 for both the ICD9 and ICD10 codes, and calculate comparability ratios (supporting Table S1). We then
recalculate comparability ratios for the 5 diseases (RESP, COPD, IHD, LC and STROKE) as the ratios of deaths for ICD9 and ICD10 (supporting Table S2). Finally, we apply these ratios to the ICD9 baseline mortality rates from 1990 to 1998.

We use archived US population in individual counties from the CDC, which are from the US Census Bureau in 1990, 2000 and 2010. The population from 1991 to 1999 was interpolated in each county between the 1990 and the 2000 censuses, and the population from 2001 to 2009 was interpolated between 2000 and 2010 (CDC 2017). The adult population above 25 yrs in
the US has steadily grown between 1990 and 2010, with an average 1.23% $yr^{-1}$ rate of increase (supporting Fig. S1).

### 2.3 The contribution of different factors to mortality trends

The overall trends in $PM_{2.5}$- and $O_3$-related mortality between 1990 and 2010 are a combination of contributions from trends in population, baseline mortality rates, and concentration. Here we separate the contributions of each factor by assuming that only a single factor was changing from 1990 to 2010, with the other two constant at 1990 levels. For example, the mortality
burden change associated with air pollution changes in year $y$ ($\Delta Mort_P^y$), relative to 1990, is calculated following Eq. (3):

$$\Delta Mort_P^y = y_0^{1990} \times AF^y \times Pop^{1990} - y_0^{1990} \times AF^{1990} \times Pop^{1990} \qquad (3)$$

Similarly, we also calculate the mortality burden change without accounting for ambient air pollution changes ($\Delta Mort_{noP}^y$) following Eq. (4):

$$\Delta Mort_{noP}^y = y_0^y \times AF^{1990} \times Pop^y - y_0^{1990} \times AF^{1990} \times Pop^{1990} \qquad (4)$$

## 3 Results

### 3.1 Air quality trends

From 1990 to 2010, annual average $PM_{2.5}$ in the model decreases significantly in the eastern US (Fig. 1-c), but slightly decreases or even increases in the northwest, southwest and west (supporting Fig. S2 and Table S3; also see supporting Fig. S3 for the US 9 regions defined by National Oceanic and Atmospheric Administration, Zhaneg et al., 2016). The dramatic
decreasing trends of $PM_{2.5}$ in the eastern US were also reported in previous studies (Gan et al., 2015; Xing et al., 2015) due



to emission reductions. The increasing trend in the west central is due in part to frequent wildfires (Dennison et al., 2014; Hand et al., 2013, 2014; Jaffe et al., 2008; Murphy et al., 2011; Spracklen et al., 2007). In general, the decadal decreasing trends in the east are larger than 2 µg m$^{-3}$ da$^{-1}$ (decade$^{-1}$) from 1990 to 2010, especially in the central (-3.48 µg m$^{-3}$ da$^{-1}$) and northeast (-3.14 µg m$^{-3}$ da$^{-1}$). The summertime average of 1hr daily maximum O$_3$ decrease significantly in the central and eastern US, generally at a rate greater than 4 ppbv da$^{-1}$. It also decreases in the western US, but at a much smaller rate than in the east, generally less than 1 ppbv da$^{-1}$ (Fig. 1-f; supporting Table S3).

In Fig.2, both the spatial average and population-weighted average (PWA) annual PM$_{2.5}$ exhibit smooth decreasing trends (Fig. 2, top): the spatial average of annual PM$_{2.5}$ has decreased by 29%, from 9.07 µg m$^{-3}$ in 1990 to 6.45 µg m$^{-3}$ in 2010, with a decadal rate of decrease of 1.1 µg m$^{-3}$ da$^{-1}$. The corresponding PWA PM$_{2.5}$ decreases by 39%, from 17.61 µg m$^{-3}$ in 1990 to 10.73 µg m$^{-3}$ in 2010, with a decadal decreasing rate of 3.2 µg m$^{-3}$ da$^{-1}$. Years with high PM$_{2.5}$, such as in 1994, 1996, and 2000, are mainly caused by increases in organic carbon due to large wildfires in the western US (Spracklen et al., 2007). Both the spatial average and PWA O$_3$ also exhibit decreasing trends over the past 2 decades, with greater inter-annual variability resulting from meteorological variability (Porter et al., 2017). The spatial average O$_3$ concentration decreases by 9%, from 55.02 ppbv in 1990 to 49.99 ppbv in 2010, decreasing at a rate of 2.4 ppbv da$^{-1}$. The PWA O$_3$ also decrease by 9%, from 58.96 ppbv in 1990 to 53.57 ppbv in 2010, decreasing at a rate of 3.0 ppbv da$^{-1}$. We also calculate the air quality and mortality burden trends separately for two 11-yr periods, 1990 to 2000 and 2000 to 2010, following Astitha et al. (2017). Both PM$_{2.5}$ and O$_3$ decrease more strongly in the second decade than in the first decade for both spatial average and PWA (Table 1), consistent with previous findings (Astitha et al., 2017; Gan et al., 2015; Porter et al., 2017; Xing et al., 2015).

We then calculate trends in the number of days annually that exceed the daily PM$_{2.5}$ standard (35µg m$^{-3}$), and the daily MDA8 O$_3$ standard (70 ppbv) (supporting Fig. S4). The exceedance days decrease for both PM$_{2.5}$ and O$_3$, especially in the eastern US. In 2010, fewer than 5 days exceed the air quality standard for the majority of the US (supporting Fig. S4, b,e). We also calculate the population exposure exceedances by multiplying the population (adults > 25 yrs old) by the number of air quality exceedance days in each grid cell. The PM$_{2.5}$ population exposure exceedances have decreased from 5340 million people-days in 1990 to 1042 million people-days in 2010, and the O$_3$ population exposure exceedances has decreased from 4691 million people-days in 1990 to 2236 million people-days in 2010 (supporting Fig. S1). These decreases in population exposure exceedances occur despite population growth over this period.

### 3.2 Mortality burdens trends and contributing factors

The mortality burdens associated with exposure to ambient PM$_{2.5}$ in the US steadily decrease by 53%, from 123,700 (70,800-178,100) deaths yr$^{-1}$ in 1990 to 58,600 (24,900-98,500) deaths yr$^{-1}$ in 2010 (Fig. 3). The leading cause for PM$_{2.5}$-related mortality is IHD, which decreases by 55%, from 96,500 (62,600-132,500) deaths yr$^{-1}$ in 1990 to 43,600 (21,500-68,700) deaths yr$^{-1}$, followed by LC which has decreased by 44%, from 12,500 (2,500-21,000) deaths yr$^{-1}$ in 1990 to 7,000 (900-13,400) deaths yr$^{-1}$ in 2010 (supporting Table. S4). The PM$_{2.5}$ mortality burden per 100,000 adults is much higher in the east than the west for both 1990 and 2010 (Fig. 4), due to the higher PM$_{2.5}$ concentrations (Fig. 1).



The PM$_{2.5}$-related mortality burden in 2010 would have decreased by only 24% (94,400 deaths yr$^{-1}$ in 2010, 95%CI, 50,300-139,800) compared with that in 1990, if the PM$_{2.5}$ concentrations had stayed constant over the period 1990-2010 (Table 1), due to decreases in the baseline mortality rates for the specific causes of death that PM$_{2.5}$ influences (Fig. 3), especially IHD (supporting Fig. S5), despite the population increase. Therefore, the reduction in PM$_{2.5}$ concentrations from 1990 to 2010

significantly accelerates the decrease in the mortality burden. The decreased PM$_{2.5}$ concentration avoided roughly 35,800 (38%) PM$_{2.5}$-related deaths in 2010, compared to the case if current air quality stays at level in 1990 (estimated as the 2010 mortality burden minus the "concentration change excluded" case in 2010). The benefit of the decreased PM$_{2.5}$ concentration could also be estimated as the "concentration change only" case in Figure 3, yielding 78,900 (35,700-129,200) deaths yr$^{-1}$ in 2010, decreasing by 36% (-44,800 deaths yr$^{-1}$) compared with 1990. The population increases from 1991 to 2010 would lead

to increases in the PM$_{2.5}$ mortality burden, but that increase is smaller than the combined reduction from decreasing PM$_{2.5}$ concentrations and baseline mortality rates (supporting Figs. S6 and S7).

When separating the two 11-yr periods, the PM$_{2.5}$-related mortality burden has decreased 45% from 2000 to 2010 (decreasing trend of -4400 deaths yr$^{-1}$), much higher than the 15% decrease from 1990 to 2000 (decreasing trend of -2100 deaths yr$^{-1}$) (supporting Table S5). The detrended annual PM$_{2.5}$-related mortality burden has a coefficient of variation (CV, standard

deviation divided by average) of 4%, mainly caused by inter-annual variation in PM$_{2.5}$ concentrations (supporting Table S6 and Fig. S7).

We also calculate burdens and trends for each state individually (Table 2). The three states with the highest PM$_{2.5}$ mortality burden in 1990 are New York (NY, 13,700 deaths yr$^{-1}$), California (CA, 9,500 deaths yr$^{-1}$) and Pennsylvania (PA, 9,200 deaths yr$^{-1}$); and in 2010, NY (5,100 deaths yr$^{-1}$), Texas (TX, 4,200 deaths yr$^{-1}$) and Ohio (OH, 3,900 deaths yr$^{-1}$). NY has seen the

largest benefits of mortality burden decreases (-8,500 deaths yr$^{-1}$), followed by CA (-6,100 deaths yr$^{-1}$) and PA (-5,500 deaths yr$^{-1}$). For the relative mortality burden changes, generally large percent decreases in PM$_{2.5}$-related mortality are seen in western, northern, and northeastern states (including Nevada, Utah, Colorado, Montana, Maine and Vermont) (Fig.5), because the PM$_{2.5}$ concentrations in 2010 are very low or even fall below the low-concentration threshold in these states (Fig. 1), as confirmed by the mortality burden changes from concentration changes alone (supporting Table S7). For other states in eastern US with

large relative mortality burden changes, the contributing factors are different. For example, for Connecticut, the relative mortality burden changes from the decrease of PM$_{2.5}$ concentration are larger than that from the decrease of the baseline mortality rates. However, for Massachusetts, NY and PA, the decreases of baseline mortality rates have a slightly larger effect than that from the decrease of PM$_{2.5}$ concentration. For CA, the effects from the decrease of baseline mortality rates and PM$_{2.5}$ concentration are comparable (supporting Table S7).

The mortality burden associated with exposure to O$_3$ from chronic respiratory disease (RESP) has increased by 13%, from 10,900 (3,700-17,500) deaths yr$^{-1}$ in 1990 to 12,300 (4,100-19,800) deaths yr$^{-1}$ in 2010 (Fig.3). The O$_3$ mortality burden per 100,000 adults is highest in the midwest and southwest (Fig.4). The O$_3$-related mortality burden in 2010 would have increased by 55% (10,600 deaths yr$^{-1}$ in 2010, 95%CI, 3,600-17,100) compared with that in 1990, if the O$_3$ concentration had stayed constant over the period 1990-2010 (Fig. 3), due to increases in both population and baseline mortality rates (supporting Fig.





S6). The decreased $O_3$ concentration would have avoided roughly 4,600 (27%) $O_3$-related deaths in 2010, compared to the case if ozone concentrations stay at level in 1990 (estimated as the 2010 mortality burden minus the "concentration change excluded" case in 2010). The benefit of the decreased $O_3$ concentration could also be estimated as the "concentration change only" case in Figure 3, yielding 8,100 (2,700-13,100) deaths $yr^{-1}$ in 2010, decreasing by 25% (-2800 deaths $yr^{-1}$) compared

with 1990. The change in $O_3$ generally reduces the mortality burden relative to 1990 with some inter-annual variation (supporting Fig. S7) due to meteorology and wildfires (Porter et al., 2017), while the increases of population and baseline mortality rates generally increase the mortality burden, with a larger contribution from the population change (supporting Fig. S7).

When separating the $O_3$ mortality trends into two decades, we find that the burdens decrease slightly (-70 deaths $yr^{-1}$) from

2000 to 2010, compared with the increasing trend from 1990 to 2000 (240 deaths $yr^{-1}$) (supporting Table S5). The increasing trend in the first decade is caused by the combined effect of increases in baseline mortality rates and population, while the decreasing trend in the second decade is dominated by decreases in $O_3$ concentration (supporting Fig. S7). The inter-annual variability for the detrended annual $O_3$ mortality burden from 1990 to 2010 (CV of 12%) is larger than $PM_{2.5}$ (CV of 4%), caused mainly by variations in $O_3$ concentrations from 1990 to 2010 (supporting Table S6).

The three states with the highest $O_3$ mortality burden in 1990 are CA (910 deaths $yr^{-1}$), Florida (FL, 740 deaths $yr^{-1}$) and NY (700 deaths $yr^{-1}$); and in 2010, CA (1270 deaths $yr^{-1}$), TX (900 deaths $yr^{-1}$) and FL (770 deaths $yr^{-1}$) (Table 2). CA has seen the largest $O_3$ mortality burden increases (360 deaths $yr^{-1}$), followed by TX (230 deaths $yr^{-1}$) and Arizona (AZ, 140 deaths $yr^{-1}$), with the greatest decrease in NY (-90 deaths $yr^{-1}$). For the relative mortality burden changes, large percent decreases in $O_3$-related mortality are seen in the northwest (Washington and Oregon) and northeast US, mainly caused by significant $O_3$

decreases (supporting Table S7), while the greatest percent increases occur in the southwest US driven mainly by large population increases, and also the baseline mortality rate increases.

Previous health impact assessments have used national baseline mortality rates (Cohen et al., 2017; Silva et al., 2016a, 2016b, etc.), but baseline mortality rates can vary strongly within individual counties (supporting Figure S5; Dwyer-Lindgren et al., 2016). We performed sensitivity analyses by applying the national average baseline mortality rates for each disease to every

county in the mortality burden calculations. We find that the $PM_{2.5}$ mortality burden calculated from the national average baseline mortality rates are lower than those calculated from the county-level baseline mortality rates, ranging among individual years from -2.2% to -1.3% (supporting Table S8). For the $O_3$ mortality burden, the difference between using the national average baseline mortality rates and our best estimates range from -1.1% to 2.0% (supporting Table S8). However, using the national average baseline mortality rates fails to capture regional mortality burden hotspots for both $PM_{2.5}$ and $O_3$

(supporting Figs. S8-S9), demonstrating the value of using county-level baseline mortality rates where possible.

**3.3 Comparison with previous studies**

The mortality burden associated with $PM_{2.5}$ calculated in our study generally aligns with several previous findings (Fig. 6; also supporting Table S9). Our $PM_{2.5}$ mortality burden is higher than that reported by Cohen et al. (2017) in 1990 (17% higher) and



1995 (4% higher), and lower in 2000 (-0.5%), 2005 (-17%) and 2010 (-30%) (Fig. 6). The overestimation of $PM_{2.5}$ mortality burdens in the early 2000s are likely due to the higher population-weighted $PM_{2.5}$ concentration simulated by WRF-CMAQ (Fig. 2), compared with Cohen et al. (2017), in which they estimated the $PM_{2.5}$ concentration based on data-fusion of air quality model outputs, satellite retrievals and ground observations. The lower mortality burdens in the second decade (from 2000 to 2010) in our study likely reflect that Cohen et al. (2017) included hemorrhagic stroke and lower respiratory infections in the $PM_{2.5}$ related mortality burden, in addition to COPD, LC, IHD and STROKE, and used an updated integrated exposure–response function. While the absolute value is similar, our results show a stronger decreasing trend (-3000 deaths $yr^{-1}$) than Cohen et al. (2017) (-1000 deaths $yr^{-1}$), which may result from the overestimation of $PM_{2.5}$ decreasing trends in our model relative to ground observations (Gan et al., 2016). The $PM_{2.5}$ mortality burdens estimated in our study are much lower than those from Fann et al. (2017), but the temporal patterns are similar, mainly because Fann et al. (2017) estimated the total all-cause mortality with a different HIF.

To compare with Cohen et al. (2017), who reported the $O_3$ mortality burden from COPD, which is a subset of RESP, we recalculate $O_3$ mortality burden from COPD (supporting Table S4). The newly calculated $O_3$ mortality burden from the COPD is generally lower than the estimate of Cohen et al. (2017) by 8%-30% (Fig. 6). This could be caused by the fact that for the $O_3$ changes, we use the summertime (April to September) average of 1-hr daily maximum, while Cohen et al. (2017) used the three-month average, which will be higher. The temporal trend for the $O_3$ mortality burdens from our study is similar with that from Cohen et al. (2017), except that the burden decreases after 2005 in our study, but increases in Cohen et al. (2017). The $O_3$ mortality burden from the RESP disease in 2005 estimated from our study is much lower than two previous studies (Fann et al., 2012; Punger and West, 2013; supporting Table S10). As discussed in the methods, the lower US background $O_3$ concentration used in these two studies (22ppb in the eastern US and 30ppb in the western US) could lead to higher $O_3$ mortality burden. We then did sensitivity analysis by using the pre-industrial $O_3$ concentration simulated by an ensemble of model outputs (supporting Fig. S10) as the counterfactual risk exposure factor, and recalculated the $O_3$ mortality burden with the RESP. The new calculated $O_3$ mortality burdens are estimated to be 64%-100% higher than current estimation from RESP using the low-concentration threshold (supporting Table S10). In Fig. 6, we see that the new estimated $O_3$ mortality burden from RESP in 2005 (dashed line) is now comparable with the two previous studies.

## 4 Conclusions

Significant improvements in air quality occurred in the US from 1990 to 2010, which we estimate to have decreased the population-weighted annual average $PM_{2.5}$ by 39%, and summertime (April to September) 1-hr daily maximum $O_3$ by 9%. However, both $PM_{2.5}$ and $O_3$ are still a great threat to the public health in US, with estimated mortality burdens of 58,600 (24,900-98,500) deaths $yr^{-1}$ and 12,300 (4,100-19,800) deaths $yr^{-1}$ in 2010, respectively. The mortality burdens associated with exposure to ambient $PM_{2.5}$ have decreased by 54% over the past two decades. However, if the annual $PM_{2.5}$ concentration levels had remained constant during 1990-2010, the associated mortality burden would have only decreased by 24%, due to





decreases in the baseline mortality rates of causes of death affected by $PM_{2.5}$ and despite population growth. The air quality improvements have significantly decreased the mortality burden, avoiding roughly 35,800 (38%) $PM_{2.5}$-related deaths in 2010, compared to the case if air quality had stayed at 1990 levels.

The mortality burdens attributable to $O_3$ are estimated to have increased by 13% during the same period. However, without the
emission reductions associated with implementation of measures under the CAA and the $NO_x$ SIP Call, the $O_3$ mortality burden would have increased by 55% during 1990-2010. In calculating the $O_3$ mortality burdens, we use the average of 1hr-daily maximum $O_3$, and the RR from Jerrett et al. (2009), but higher $O_3$ mortality burdens would likely have resulted had we used RRs from Turner et al. (2016). We estimate that the air quality improvements have avoided 4,600 (27%) $O_3$-related deaths in 2010, compared to the case if air quality had stayed at 1990 levels.

We also estimate the inter-annual variability in mortality burdens considering air pollutant concentrations in individual years and annual county-level baseline mortality rates, and find that the $O_3$ mortality burdens are more variable (CV of 12%) than for $PM_{2.5}$ (CV of 4%), mainly because of inter-annual variability in concentrations.

The uncertainties in air pollution-related mortality estimates presented in this study are based on the uncertainty in relative risks for the specific causes of death only, and do not account for uncertainties in population and baseline mortality rates
(which are likely small), nor for uncertainty in the modeled air pollutant concentration. Uncertainties also exist due to the assumption of equal toxicity for different components of $PM_{2.5}$ (Li et al., 2015). For our analysis, we use modeled air pollutant concentration without any bias-correction based on either in-situ observation or satellite data (Brauer et al., 2015; Hogrefe et al., 2009; van Donkelaar et al., 2015; Xu et al., 2016). In our study, the $PM_{2.5}$ mortality burdens trend may be overestimated, and $O_3$ mortality burdens underestimated, based on comparing the modeled air pollution trends with the observations. Despite
these uncertainties, this study illustrates the importance of past air pollutant reductions for public health in the US, and of continued air pollution controls to reduce air pollution-related mortality.

**Data availability**: The 21-yrs model outputs for the coupled WRF-CMAQ model, as well as the annual county level baseline mortality rates can be obtained by contacting the corresponding author (Y. Zhang, yuqiangzhang.thu@gmail.com,
Yuqiang.zhang@duke.edu).

**Competing interests**. The authors declare that they have no conflicts of interest.

**Acknowledgement:** This research was supported in part by an appointment to the Research Participation Program at the
U.S. EPA, Office of Research and Development (ORD), administered by the Oak Ridge Institute for Science and Education (ORISE) through an interagency agreement between the U.S. Department of Energy and the U.S. EPA, and also in part by National Aeronautics and Space Administration Health and Air Quality Applied Science Team #NNX16AQ80G. This work was also supported in part by China Ministry of Science and Technology National Key R & D program (2016YFC0207601). We thank Raquel Silva and Neal Fann from U.S. EPA for useful discussions.  We greatly acknowledge Ana Rappold and





Geoffrey Peterson from the U.S. EPA for their comments and suggestions on the initial version of this manuscript. We also appreciate the free usage of the county level baseline mortality and population data from US Centers for Disease Control and Prevention (CDC).

5    **Disclaimer**: Although this work has been reviewed and approved for publication by the U.S. EPA, the views expressed in this paper are those of the authors and do not necessarily represent the views or policies of the U.S. EPA.

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

|  |  | 2010 (deaths yr$^{-1}$) | Relative Changes |
|---|---|---|---|
| **PM$_{2.5}$** | Mortality burden | 58,600 (24,900-98,500) | -54% |
|  | Concentration change only | 78,900 (35,700-129,200) | -36% |
|  | MortalityRates change only | 68,300 (35,800-101,300) | -45% |
|  | Population change only | 173,500 (99,900-250,000) | 40% |
|  | Concentration change excluded | 94,400 (50,300-140,000) | -24% |
| **O$_3$** | Mortality burden | 12,300 (4,100-19,800) | 13% |
|  | Concentration change only | 8,100 (2,700-13,100) | -25% |
|  | MortalityRates change only | 13,100 (4,400-21,000) | 20% |
|  | Population change only | 14,100 (4,800-22,700) | 30% |
|  | Concentration change excluded | 16,900 (5,700-27,000) | 55% |



**Table 2. The mortality burden for 48 US states and the District of Columbia in 1990 and 2010, and the absolute changes from 1990 to 2010. Units are deaths yr$^{-1}$.**

| States | PM$_{2.5}$-related mortality | | | O$_3$-related mortality | | |
|---|---|---|---|---|---|---|
| | **1990** | **2010** | **Diff** | **1990** | **2010** | **Diff** |
| AL | 2135 | 1166 | -969 | 159 | 238 | -12 |
| AR | 1127 | 752 | -375 | 74 | 133 | 22 |
| AZ | 554 | 196 | -358 | 125 | 329 | 138 |
| CA | 9515 | 3420 | -6095 | 567 | 1272 | 359 |
| CO | 222 | 35 | -187 | 115 | 230 | 64 |
| CT | 1795 | 458 | -1337 | 93 | 129 | -22 |
| DC | 250 | 157 | -92 | 12 | 21 | -6 |
| DE | 492 | 264 | -227 | 26 | 54 | 14 |
| FL | 4688 | 2441 | -2246 | 483 | 774 | 34 |
| GA | 3149 | 1954 | -1195 | 221 | 413 | 51 |
| IA | 1500 | 756 | -743 | 74 | 102 | 1 |
| ID | 174 | 120 | -54 | 18 | 39 | 14 |
| IL | 7770 | 3547 | -4223 | 280 | 500 | 38 |
| IN | 3821 | 2067 | -1754 | 198 | 360 | 71 |
| KS | 1064 | 697 | -367 | 84 | 147 | 26 |
| KY | 2420 | 1388 | -1032 | 160 | 257 | 23 |
| LA | 1752 | 855 | -898 | 109 | 195 | 4 |
| MA | 3417 | 1107 | -2310 | 153 | 197 | -57 |
| MD | 2893 | 1713 | -1180 | 155 | 261 | 1 |
| ME | 347 | 5 | -341 | 28 | 21 | -19 |
| MI | 5894 | 2590 | -3304 | 220 | 407 | 46 |
| MN | 1626 | 699 | -927 | 61 | 107 | 16 |
| MO | 3135 | 1906 | -1229 | 175 | 286 | 31 |
| MS | 1352 | 608 | -743 | 75 | 124 | 6 |
| MT | 9 | 2 | -7 | 12 | 19 | 1 |
| NC | 3321 | 1961 | -1361 | 208 | 430 | 70 |
| ND | 75 | 23 | -52 | 8 | 12 | -1 |
| NE | 535 | 257 | -278 | 55 | 81 | 7 |
| NH | 453 | 73 | -380 | 24 | 25 | -13 |
| NJ | 5332 | 2196 | -3137 | 223 | 404 | 28 |
| NM | 245 | 180 | -65 | 37 | 109 | 42 |
| NV | 10 | 0 | -10 | 52 | 138 | 60 |
| NY | 13712 | 5239 | -8473 | 406 | 613 | -88 |
| OH | 7876 | 3932 | -3944 | 400 | 690 | 103 |
| OK | 1499 | 1058 | -441 | 120 | 248 | 77 |
| OR | 633 | 219 | -413 | 39 | 42 | -15 |
| PA | 9238 | 3727 | -5511 | 393 | 584 | -70 |
| RI | 630 | 172 | -457 | 32 | 44 | -5 |
| SC | 1673 | 974 | -699 | 109 | 218 | 30 |
| SD | 140 | 68 | -72 | 14 | 23 | 3 |
| TN | 3097 | 1895 | -1202 | 199 | 317 | 13 |
| TX | 6499 | 4178 | -2321 | 417 | 896 | 228 |
| UT | 107 | 10 | -96 | 25 | 72 | 29 |



| VA | 2806 | 1592 | -1214 | 183 | 336 | 29 |
|----|------|------|-------|-----|-----|-----|
| VT | 196 | 14 | -182 | 11 | 8 | -9 |
| WA | 917 | 394 | -522 | 71 | 75 | -27 |
| WI | 2479 | 977 | -1503 | 75 | 148 | 35 |
| WV | 1161 | 534 | -627 | 84 | 122 | -9 |
| WY | 1.2 | 0.4 | -0.8 | 12 | 25 | 8 |





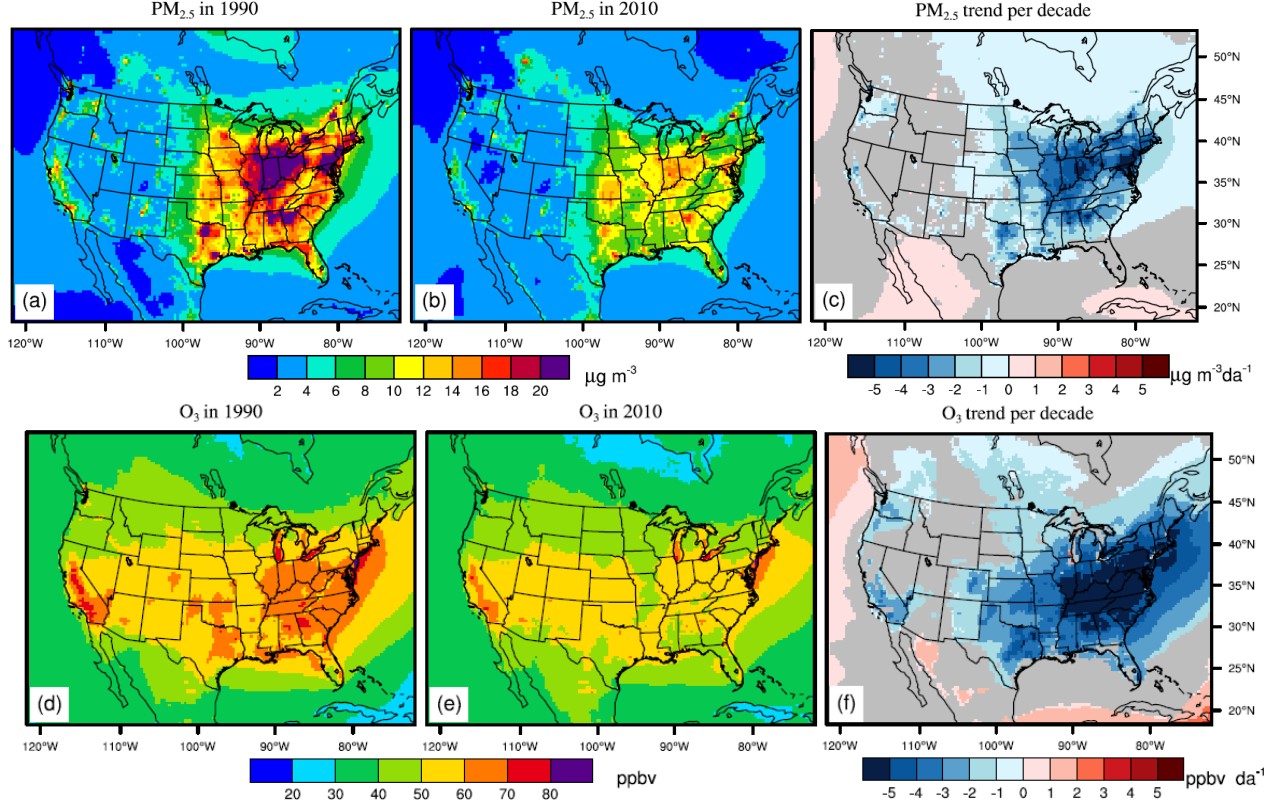

**Figure 1: Annual mean PM$_{2.5}$ (µg m$^{-3}$) in 1990 (a), 2010 (b), and the 21-yr trends (c, µg m$^{-3}$ da$^{-1}$ (µg m$^{-3}$ per decade)), and summertime average of 1hr daily maximum O$_3$ in 1990 (ppbv) (d), 2010 (e) and the trend (f, ppbv da$^{-1}$ (ppbv per decade)). The grey shaded areas in panels c and f indicate trends that are insignificant with p-values for the standard Student-t test larger than 0.05.**





**Figure 2: Population-weighted average (Popweighted-Avg) and spatial average over CONUS land areas of annual average PM₂.₅ (top) and summertime average of 1hr daily maximum O₃ (bottom) concentration from 1990 to 2010. Population-weighted average concentrations are based on population in each year. Using the same population in each year yields estimates of population-weighted concentrations that are only slightly different (not shown).**





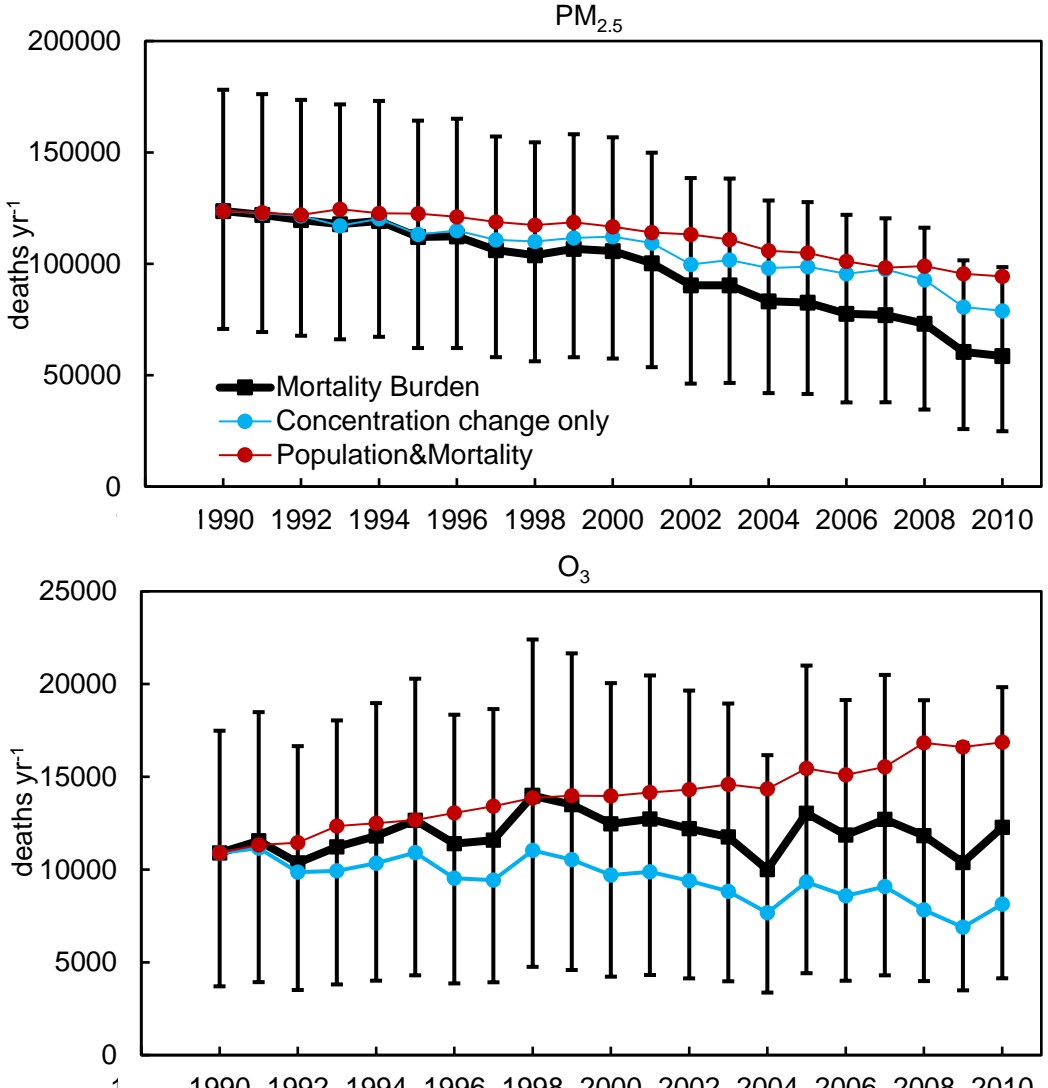

**Figure 3: Trends in the total mortality burden (black) for PM₂.₅ (top, as a total of ischemic heart disease (IHD) + stroke (STROKE) + chronic obstructive pulmonary disease (COPD) + lung cancer (LC)) and O₃ (bottom, chronic respiratory disease (RESP)), and mortality burdens considering the air quality change only (blue), and with air quality changes excluded (red). Units are deaths yr⁻¹. The error bars are the 95% CI for the total mortality burden (black).**





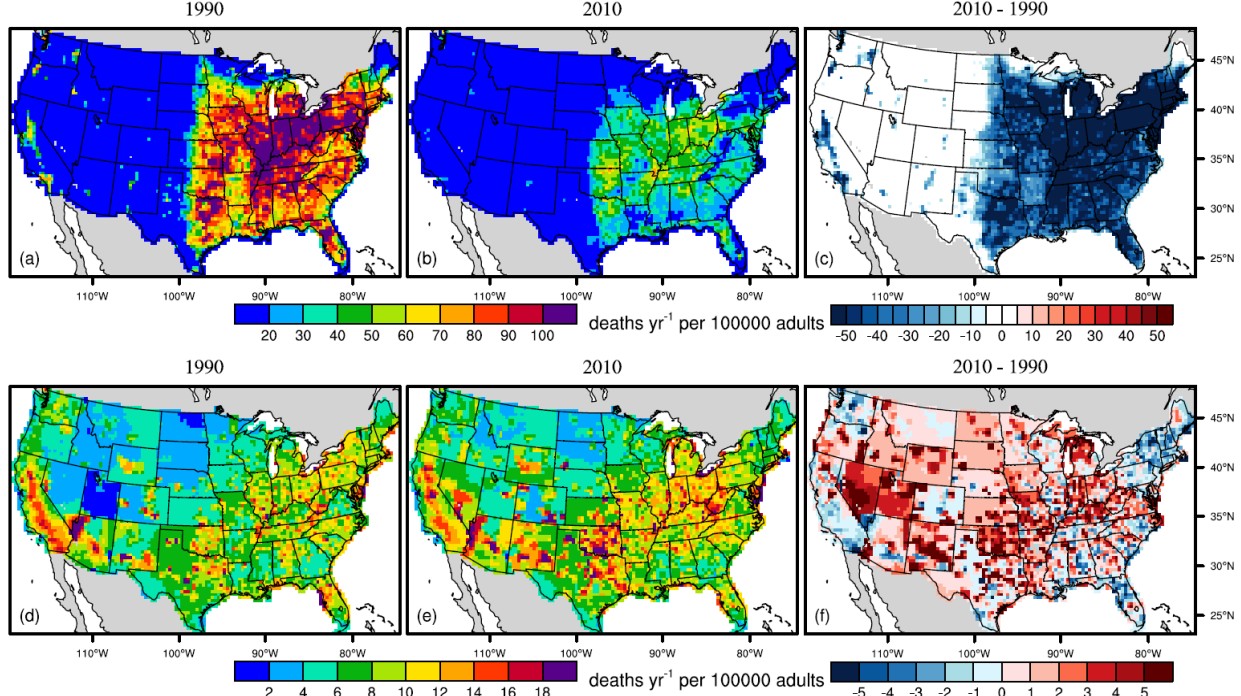

**Figure 4: The mortality burdens associated with PM$_{2.5}$ (a, b), O$_3$ (d, e) in 1990 (a, c) and 2010 (b, d), and the differences (2010 minus 1990) (c, f) for each 36km×36km grid cell. Units are deaths yr$^{-1}$ per 100,000 adults (above 25 yrs old).**





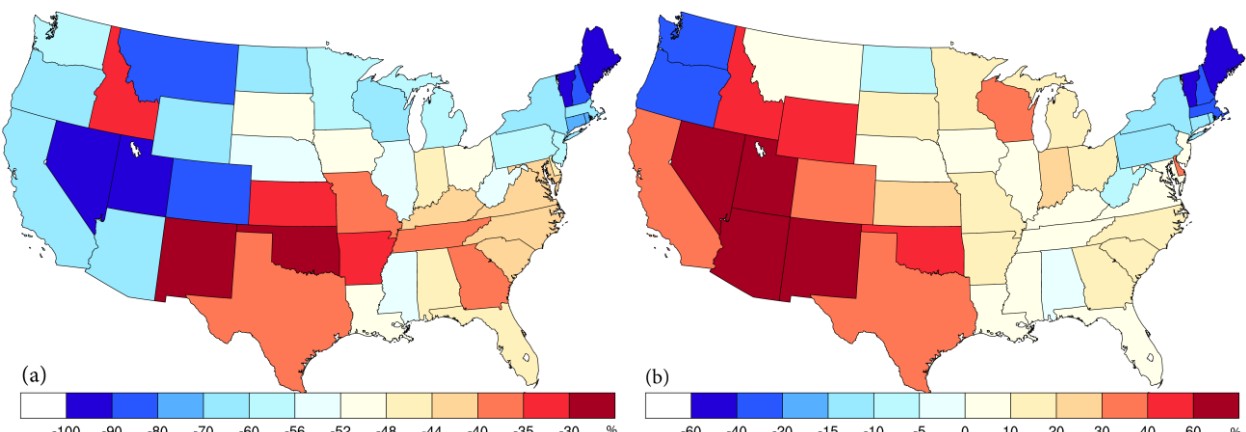

**Figure 5: Relative mortality burden changes from 1990 to 2010 for the 48 states and the District of Columbia for PM$_{2.5}$ (a) and O$_3$**

**(b). The relative changes are calculated as (2010-1990)/1990×100%. Note the different color scales for the two plots. The values for**

**the District of Columbia are -37% for PM$_{2.5}$ and -23% for O$_3$.**



**Figure 6:** Comparisons of the U.S. mortality burdens attributed to PM$_{2.5}$ (a), and O$_3$ (b) in this study, with Cohen et al., (2017),
Fann et al., (2017), Fann et al., (2012), Punger and West (2013), and Giannadaki et al., (2017). The black line for O$_3$ is the
recalculated O$_3$ mortality burden from COPD, and the black dashed line is the recalculated O$_3$ mortality burden from RESP using
the pre-industrial O$_3$ concentration as the counterfactual risk exposure factor. The error bars show the 95% CI from the RRs,
shown for this study and Cohen et al., (2017).