# Peer review of "Long-term trends in the ambient PM2.5- and O3-related mortality burdens in the United States under emission reductions from 1990 to 2010"

_Atmospheric Chemistry and Physics, 2018_

## Referee Comment (RC1) · Anonymous Referee #1 · 17 Jul 2018

General comments: This is an interesting and useful contribution which evaluates the contribution of underlying factors to long-term air pollution-related mortality trends in the continental US, with the aim of highlighting the importance of concentration reductions. This appears to be the first application of a multi-decadal air quality modeling exercise to analyse such a question.

The manuscript is clearly written, with balanced arguments and inclusion of recent, relevant literature. The methods employed are appropriate, clearly described, and well supported.

It would be helpful to have additional information on the accuracy of the linear interpo-

lation method for population across Census years. Can they discuss the accuracy of this, perhaps with reference to sources that estimate inter-Census population? There are a variety of sources for this, with more sophisticated methods than linear interpolation that rely, e.g., on the American Community Survey. Geolytics Inc. has annual products for population, or perhaps LandScan Global population.

Specific comments: Page 10, Lines 1-3: The authors are careful to talk about "reduced mortality burden", but here they mention 'avoided deaths'. Perhaps add "premature" in front of deaths.

Page 7, line 1: recommend introducing Table 1 here, since this seems to be the first time its results are mentioned.

Figure 5(a) color bars took a moment to interpret, since for (b) using symmetric saturation with cool = reduction and warm increase, but for (a) these are all reductions. Despite the note, it still took a moment. Perhaps consider only using cool colors for 5(a)?

Figure S1 – can you add a legend, perhaps, and/or indicate color of population increase line in caption?

Technical corrections: Page 6 Line 18 - refers to the split decadal trend, which is in Table S5 not Table 1. Page 6 Line 28: Table 1 says 54% not 53% Page 5 Line 29: "Zhang" instead of Zhaneg

---

## Referee Comment (RC2) · Anonymous Referee #2 · 9 Aug 2018

The manuscript by Zhang et al. examines trends in premature mortalities associated with exposure to ambient PM2.5 and O3 in the US over 3 decades. The work is valuable towards understanding the different factors driving these trends and in contributing to a body of evidence documenting the public health benefits of air pollution controls. I have some questions and suggestions that relate to how the authors present uncertainty in their analysis, how they distinguish their work from previous studies, and how they present the trends in different components of the premature mortality calculations (i.e. concentrations vs baseline mortality rates). Addressing these will amount to minor revisions, after which I believe this manuscript will be suitable for publication in ACP.

[Figure]

Major comments:

1.26 and throughout: These confidence intervals only account for a subset of the uncertainties inherent in these estimates (i.e., they ignore any inaccuracies in the air pollution model). Thus, it should be clearly stated up front what these ranges do and do not represent. The same comment applies to other places where these numbers are prominently presented, such as e.g. Table 1.

2.25 - 3.6: I'm not sure I appreciate the significance of the differences between the work in this manuscript and the previous works of Cohen et al. (2017) and Fann et al. (2017), who estimated trends in premature mortalities associated with PM2.5 (and O3 – Cohen) in the US since 1980. Yes, their analysis was only once every 5 years, not every year, but does that really make a big difference in the overall conclusions? I'm not sure what the importance of studying successive years is, or interannual variability. The authors results shown in Fig 3 would seem to indicate the answer is "not much", at least for PM2.5. I further think works such as Fann 2017 also do discuss different drivers of the trends (mortality rates, population. . .). So, I suggest the authors could go into more detail here about what these previous studies found, including their quantitative results, and also how the present work goes beyond these previous studies methodologically. Update: upon reading section 3.3 (Comparison to previous studies), I'm more informed about how these results differ. Yet still, it would probably benefit the authors to put some more of this content up front in the introduction for motivation. At the very least, Section 3.3 could be alluded to in outlining the contents of what is to come (3.7-12).

General: I have some confusion about how to separately interpret the impacts of changing concentrations from changing mortality rates. At present the manuscript seems to imply that changes in baseline mortality rates are not benefits of improving air pollution. However, if reductions in concentrations improve air quality, wouldn't this lead to reductions in mortality rates? To what extent are the evolution of these two terms in the health impact function separate? Could the authors comment on an explain this a bit more?

[Figure]

4.23: What was the basis for using an average value for the threshold? What impacts does this value have on the overall findings, quantitatively and qualitatively?

4.27: Again, what is the basis for this, and how does it affect the findings?

Results: Please provide as separate figures (there is plenty of room for this within the main body of an ACP paper): (1) trends in baseline mortality rates, (2) trends in concentrations, (3) trends in AF, (4) trends in mortalities. Providing these pieces of information separately would make parsing the results of this paper so much more straightforward. At less than 10 pages of text and only 6 figures, presently, there's no reason to place any of this in the SI.

Fig 5: I strongly object to the choice of color scale for panel (a). The potential for misunderstanding the results is quite high. Please use red-blue colors scales as commonly understood, which blue being negative and red being positive. Or pick an entirely different color scheme for these results that are strictly negative.

Minor comments and corrections:

18: Perhaps, more precisely, "exposure to these pollutants are associated with. . ."

abstract: From the perspective of air quality control and the audience of ACP, it seems more interesting to report how many premature mortalities would have been avoided by PM2.5 and O3 reductions in the absence of changes in baseline mortality rates (rather than the other way around). That being said, perhaps these are the numbers that are reported in the last few lines of the abstract? It's not clear if these are / are not accounting for changes in mortality rates (or population).

2.15: observations sites –> observations

2.20: it's note entirely clear how the authors are separating the benefits of the NAAQS from those of the technologies put in place to meet the NAAQS – seems like these are perhaps two sides of the same coin.

2.24: I understand the wording, but if one accept the health impact analysis framework used here, then does it matters not where people live as much as where they die?

2.15 - 2.24: Note the difference in tone between the assertiveness of this work ("improvements were mainly driven by ambient air quality standards...", and that of Fann 2017 who state "it is difficult to attribute this reduction to specific policy interventions....many factors are likely to have contributed...federal air quality policies are likely to have played an important role". It then seems that different federal regulations are cited, such as Acid Rain program. Thoughts about why the present work is a bit more sure of the role of regulations in this regard? Were EPA authors just being more cautious with their wording for professional reasons?

4.16: I doubt this is what the authors meant to say. If RRs were downloaded from the GBD web site, then there would be no role for the simulations of O3 and PM2.5 concentrations described in section 2.1.

4.25: The justification here seems a bit odd, as if the authors decided that sticking with the mismatched population age-ranges from their previous work would take precedent over correctly matching age ranges with the epidemiological study of Jerrett 2009.

4.28 - 5.15: Would it be possible to avoid these types of inconsistencies in reporting to use heath impact functions associated with all-cause rather than cause-specific mortality rates, even if the epidemiological evidence of the responses isn't as robust?

7.24: in the eastern

Title: should indicate this is about ambient AQ? Or that is obvious?

---

## Author Comment (AC1) · 21 Sep 2018

Response to review #1 on acp-2018-498

Long-term trends in the PM$_{2.5}$- and O$_3$-related mortality burdens in the United States under emission reductions from 1990 to 2010

Yuqiang Zhang, J. Jason West, Rohit Mathur, Jia Xing, Christian Hogrefe, Shawn J. Roselle, Jesse O. Bash, Jonathan E. Pleim, Chuen-Meei Gan, David C. Wong

General comments: This is an interesting and useful contribution which evaluates the contribution of underlying factors to long-term air pollution-related mortality trends in the continental US, with the aim of highlighting the importance of concentration reductions. This appears to be the first application of a multi-decadal air quality modeling exercise to analyse such a question.
The manuscript is clearly written, with balanced arguments and inclusion of recent, relevant literature. The methods employed are appropriate, clearly described, and well supported.

We thank referee #1 for the very positive comments on our manuscript. We also appreciate the reviewer for the constructive suggestions, which have helped us improve the manuscript. All referee comments (in blue below) have been carefully addressed, and changes incorporated in the revised manuscript are shown using the track-changes option.

It would be helpful to have additional information on the accuracy of the linear interpolation method for population across Census years. Can they discuss the accuracy of this, perhaps with reference to sources that estimate inter-Census population? There are a variety of sources for this, with more sophisticated methods than linear interpolation that rely, e.g., on the American Community Survey. Geolytics Inc. has annual products for population, or perhaps LandScan Global population.
**Response:** The interannaul population between two censuses (1990 and 2000; 2000 and 2010) was not directly linear interpolated by us, instead they were derived from the Population Estimates project by US Census Bureau. The intercensal population estimates are estimates made for the years between two completed censuses which take into account the census at both the beginning and end of the decade.
(https://www.cdc.gov/nchs/data/nvss/bridged_race/Documentation_bridge_postcenv2017.pdf, accessed 5 September 2018).

From their documentation, the population products from the American Community Survey of Geolytics Inc. and the LandScan Global population both adopted the annual mid-year national population estimates from the US Bureau of Census (https://landscan.ornl.gov/documentation, accessed 5 September 2018) (http://www.geolytics.com/USCensus,AmericanCommunitySurvey(ACS),Data,Features,Products.asp, accessed 5 September 2018).

To avoid confusion, we rewrote the sentence in line 17-19 in page 5 (page and line numbers are in the revised manuscript):
"Annual population in the US at county level was taken from the US Bureau of Census, which reported populations associated with the 1990, 2000, and 2010 censuses and estimated population for each year in between (CDC 2017;

https://www.cdc.gov/nchs/data/nvss/bridged_race/Documentation_bridge_postcenv2017.pdf, accessed 5 September 2018)"

Specific comments: Page 10, Lines 1-3: The authors are careful to talk about "reduced mortality burden", but here they mention 'avoided deaths'. Perhaps add "premature" in front of deaths.

**Response:** We thank the reviewer for the suggestion. In the revised manuscript, we add the word "premature" as the referee suggested.

"The air quality improvements have significantly decreased the mortality burden, avoiding roughly 35,800 (38%) $PM_{2.5}$-related premature deaths in 2010, compared to the case if air quality had stayed at 1990 levels."

Page 7, line 1: recommend introducing Table 1 here, since this seems to be the first time its results are mentioned.

**Response:** We add the introduction to Table 1 in the beginning of this paragraph, and also rewrite the first sentence:

"Table 1 shows the mortality burdens for $PM_{2.5}$ and $O_3$ in 2010, and also the burden changes since 1990 from different contributing factors. From the table, we see that the $PM_{2.5}$-related mortality burden in 2010 would have decreased by only 24% (94,400 deaths $yr^{-1}$ in 2010, 95%CI, 50,300-139,800) compared with that in 1990, if the $PM_{2.5}$ concentrations had stayed constant over the period 1990-2010,"

Figure 5(a) color bars took a moment to interpret, since for (b) using symmetric saturation with cool = reduction and warm increase, but for (a) these are all reductions. Despite the note, it still took a moment. Perhaps consider only using cool colors for 5(a)?

**Response:** This figure is now Figure 6, after we added a new figure 5 to the paper. We have changed panel a of this figure to use only cool colors in different shades, as suggested by the reviewer. Thank you for this very good suggestion.

Figure S1 – can you add a legend, perhaps, and/or indicate color of population increase line in caption?

**Response:** We now add a legend in Figure S1, and also in the caption, we wrote:

"The red line is the US total adult population > 25 yrs old from 1990 to 2010 with the y-axis on the right."

Technical corrections: Page 6 Line 18 - refers to the split decadal trend, which is in Table S5 not Table 1.

**Response:** We thank the reviewer for noticing this. We now changed "Table 1" to "supporting Table S5".

Page 6 Line 28: Table 1 says 54% not 53%

**Response:** The reviewer is correct. We have updated the number in the revised manuscript.

Page 5 Line 29: "Zhang" instead of Zhaneg.

**Response:** We thank the reviewer for pointing this out. We now made the changes in the revised manuscript.

---

## Author Comment (AC2) · 21 Sep 2018

Response to review #2 on acp-2018-498

Long-term trends in the $PM_{2.5}$- and $O_3$-related mortality burdens in the United States under emission reductions from 1990 to 2010

Yuqiang Zhang, J. Jason West, Rohit Mathur, Jia Xing, Christian Hogrefe, Shawn J. Roselle, Jesse O. Bash, Jonathan E. Pleim, Chuen-Meei Gan, David C. Wong

The manuscript by Zhang et al. examines trends in premature mortalities associated with exposure to ambient $PM_{2.5}$ and $O_3$ in the US over 3 decades. The work is valuable towards understanding the different factors driving these trends and in contributing to a body of evidence documenting the public health benefits of air pollution controls. I have some questions and suggestions that relate to how the authors present uncertainty in their analysis, how they distinguish their work from previous studies, and how they present the trends in different components of the premature mortality calculations (i.e. concentrations vs baseline mortality rates). Addressing these will amount to minor revisions, after which I believe this manuscript will be suitable for publication in ACP.

**Response**: We thank referee #2 for the positive and constructive suggestions, which have helped us improve the manuscript. We have responded to each comment below and have noted the page and line number for each revision to the manuscript. (blue colors are for referee's comments).

Major comments:
1.26 and throughout: These confidence intervals only account for a subset of the uncertainties inherent in these estimates (i.e., they ignore any inaccuracies in the air pollution model). Thus, it should be clearly stated up front what these ranges do and do not represent. The same comment applies to other places where these numbers are prominently presented, such as e.g. Table 1.

**Response:** The reviewer is right that the confidence intervals we reported only consider the uncertainty from the RRs. We rephrase the sentence in page 4 line 28-29 (the pages and the numbers are referring the new draft) to clarify this:

"Uncertainties in air pollution-related mortality burden calculations are based on the uncertainty in RRs only, ignoring those in modeled air pollutant concentrations, and population and baseline mortality rates, which may be larger than that from the RRs but not in the scope of our study."

We also clarify the uncertainties in page 7 line 3 when we first report the mortality burdens:
"The mortality burdens associated with exposure to ambient $PM_{2.5}$ in the US steadily decrease by 54%, from 123,700 (95% confidence interval considering the uncertainty in relative risk only, 70,800-178,100) deaths yr-1 in 1990 to 58,600 (24,900-98,500) deaths yr-1 in 2010 (Fig. 3)."

As reported in several recent studies (Silva et al., 2016a; Liang et al., 2018), the uncertainties from the modeled air pollutants concentration may be greater than those from RRs and baseline mortality rates. So we suggest that ensembles of air quality models be used to quantify uncertainties where plausible. We then add the following sentence into the discussion at page 10 line 15:

"Previous studies have shown that the uncertainties from the modeled air pollutants concentrations may be greater than uncertainties in baseline mortality or relative risk, so the use of model ensembles is suggested to better quantify the uncertainty (Silva et al., 2016a; Liang et al., 2018)."

**References:**
Silva, R. A., West, J. J., Lamarque, J. F., Shindell, D. T., Collins, W. J., Dalsoren, S., Faluvegi, G., Folberth, G., Horowitz, L. W., Nagashima, T., Naik, V., Rumbold, S. T., Sudo, K., Takemura, T., Bergmann, D., Cameron-Smith, P., Cionni, I., Doherty, R. M., Eyring, V., Josse, B., MacKenzie, I. A., Plummer, D., Righi, M., Stevenson, D. S., Strode, S., Szopa, S. and Zengast, G.: The effect of future ambient air pollution on human premature mortality to 2100 using output from the ACCMIP model ensemble, Atmos. Chem. Phys., 16(15), 9847–9862, doi:10.5194/acp-16-9847-2016, 2016a.

Liang, C. K., West, J. J., Silva, R. A., Bian, H., Chin, M., Davila, Y., Dentener, F. J., Emmons, L., Flemming, J., Folberth, G., Henze, D., Im, U., Jonson, J. E., Keating, T. J., Kucsera, T., Lenzen, A., Lin, M., Tronstad Lund, M., Pan, X., Park, R. J., Pierce, R. B., Sekiya, T., Sudo, K. and Takemura, T.: HTAP2 multi-model estimates of premature human mortality due to intercontinental transport of air pollution and emission sectors, Atmos. Chem. Phys., 18(14), 10497–10520, doi:10.5194/acp-18-10497-2018, 2018.

2.25 - 3.6: I'm not sure I appreciate the significance of the differences between the work in this manuscript and the previous works of Cohen et al. (2017) and Fann et al. (2017), who estimated trends in premature mortalities associated with $PM_{2.5}$ (and $O_3$– Cohen) in the US since 1980. Yes, their analysis was only once every 5 years, not every year, but does that really make a big difference in the overall conclusions? I'm not sure what the importance of studying successive years is, or interannual variability. The authors results shown in Fig 3 would seem to indicate the answer is "not much", at least for $PM_{2.5}$. I further think works such as Fann 2017 also do discuss different drivers of the trends (mortality rates, population: : :). So, I suggest the authors could go into more detail here about what these previous studies found, including their quantitative results, and also how the present work goes beyond these previous studies methodologically. Update: upon reading section 3.3 (Comparison to previous studies), I'm more informed about how these results differ. Yet still, it would probably benefit the authors to put some more of this content up front in the introduction for motivation. At the very least, Section 3.3 could be alluded to in outlining the contents of what is to come (3.7-12).

**Response:** We thank the reviewer for pointing this out. The interannual variability analysis in our study indicates whether the mortality burden at a given year, such as the results from Cohen et al., 2017 and Fann et al., 2017, is representative of years around it. Indeed we found that the interannual varability is small for $PM_{2.5}$, but larger for ozone, which we are not aware has been shown previously. In addition, after carefully reading Fann et al., 2017, we do not think they provided the drivers analysis, which provides another innovation of our study.

We add one sentence in page 3 line 5:
"The interannual variability analysis indicates whether the mortality burden at a given year, such as the results from Cohen et al. (2017) and Fann et al. (2017), is representative of years around it."

We then also add one sentence into the discussion on page 10 line 21-22:
"We found that the interannual varability is small for $PM_{2.5}$, but larger for ozone, which we are not aware has been shown previously."

General: I have some confusion about how to separately interpret the impacts of changing concentrations from changing mortality rates. At present the manuscript seems to imply that changes in baseline mortality rates are not benefits of improving air pollution. However, if reductions in concentrations improve air quality, wouldn't this lead to reductions in mortality rates? To what extent are the evolution of these two terms in the health impact function separate? Could the authors comment on an explain this a bit more?

**Response:** We agree with the reviewer that improving air quality could help to reduce the baseline mortality rates (Correia, et al., 2013; Pope et al., 2009), but the changes in baseline mortality rates were more caused by other factors such as changes in social-economic development, heathcare expenditures, and so on, than the air pollution level. The baseline mortality rates for lung cancer, ischemic heart disease and stroke decreased from 1990 to 2010, as a result of the exposed population becoming more resilient and healthier, and less susceptible to the risk of air pollution, while baseline mortality for chronic obstructive pulmonary and respiratory disease increased at the same time (see Figure S5 in our supporting materials).

To clarify this, we add the following sentence in page 4 line 31-32:
"We use baseline mortality rates from each year to calculate deaths from air pollution in each year, as changes in baseline mortality rates from other socioeconomic determinants are likely more important than changes in deaths from air pollution."

**Reference:**
Correia, A. W., Arden Pope, C., Dockery, D. W., Wang, Y., Ezzati, M. and Dominici, F.: Effect of air pollution control on life expectancy in the United States: An analysis of 545 U.S. Counties for the period from 2000 to 2007, Epidemiology, 24(1), 23–31, doi:10.1097/EDE.0b013e3182770237, 2013.

Pope, C. A., Ezzati, M. and Dockery, D. W.: Fine-Particulate Air Pollution and Life Expectancy in the United States, N. Engl. J. Med., 360(4), 376–386, doi:10.1056/NEJMsa0805646, 2009.

4.23: What was the basis for using an average value for the threshold? What impacts does this value have on the overall findings, quantitatively and qualitatively?

**Response:** The counterfactual concentration of 37.6 ppb for $O_3$ premature mortality calculations was used in the recent Global Burden Disease estimates (Cohen et al., 2017). We used this value for consistency with the GBD. Adopting modeled pre-industrial $O_3$ instead of the counterfactual concentration could lead to higher estimates of $O_3$ mortality burden at present day, as discussed in Lelieveld et al. (2013, 2015), and also Table S10 in our supporting materials.

To avoid confusion, we rewrite the sentence in page 4 line 25:
"The counterfactual concentration of 37.6 ppbv (Lim et al., 2012; Lelieveld et al., 2015) is used in our study, to be comparable with Cohen et al. (2017).

**Reference:**

Lelieveld, J., Barlas, C., Giannadaki, D. and Pozzer, A.: Model calculated global, regional and megacity premature mortality due to air pollution, Atmos. Chem. Phys., 13(14), 7023–7037, doi:10.5194/acp-13-7023-2013, 2013.

Lelieveld, J., Evans, J. S., Fnais, M., Giannadaki, D. and Pozzer, A.: The contribution of outdoor air pollution sources to premature mortality on a global scale, Nature, 525(7569), 367–71, doi:10.1038/nature15371, 2015.

4.27: Again, what is the basis for this, and how does it affect the findings?

**Response:** We used one single model results for the mortality burden estimations, and so we could not include uncertainties from the modeled air pollution without further assumptions about those uncertainties. However, from previous findings, the uncertainties from the modeled air pollutants concentration are likely larger than larger than those from RRs and baseline mortality rates. So we suggest that ensemble air quality model results could be used to quantify the overall uncertainties. We now add discussion of this at the end of the manuscript in page 10 line 25-27:

"Previous studies have shown that the uncertainties from the modeled air pollutants concentrations may be greater than uncertainties in baseline mortality or relative risk, so the use of model ensembles is suggested to better quantify the uncertainty (Silva et al., 2016a; Liang et al., 2018)."

Results: Please provide as separate figures (there is plenty of room for this within the main body of an ACP paper): (1) trends in baseline mortality rates, (2) trends in concentrations, (3) trends in AF, (4) trends in mortalities. Providing these pieces of information separately would make parsing the results of this paper so much more straightforward. At less than 10 pages of text and only 6 figures, presently, there's no reason to place any of this in the SI.

**Response:** We thank the reviewer for the suggestion. We now move Figure S5 (the trends for baseline mortality rates) from the supporting materials into the main text as a new Figure 5. The numbers of the other figures in the main text as well as in the supporting are all updated as well. Trends in concentration are shown in Figure 2, trends in AF will be similar as trends in concentration, and trends in deaths are our main results in Figures 3 and 4.

Fig 5: I strongly object to the choice of color scale for panel (a). The potential for misunderstanding the results is quite high. Please use red-blue colors scales as commonly understood, which blue being negative and red being positive. Or pick an entirely different color scheme for these results that are strictly negative.

**Response:** We thank the reviewer for pointing this out. Now we updated the plot by using blue meaning negative only. See Figure 6 in the new manuscript.

Minor comments and corrections:

18: Perhaps, more precisely, "exposure to these pollutants are associated with…."

**Response:** We now rewrite this sentence:

"Exposure to these air pollutants is associated with premature death."

abstract: From the perspective of air quality control and the audience of ACP, it seems more interesting to report how many premature mortalities would have been avoided by PM2.5 and O3 reductions in the absence of changes in baseline mortality rates (rather than the other way around). That being said, perhaps these are the numbers that are reported in the last few lines of the abstract? It's not clear if these are / are not

accounting for changes in mortality rates (or population).

**Response:** The numbers reported in page 2 line 1-3 in the abstract were calculated using the mortality burden in 2010 considering all three factors (concentration, mortality rates and population) minus the mortality burden in 2010 considering two factors only (mortality rates and population; also see Figure 3 the black and red line). So here we are reporting the premature deaths which would have been avoided by $PM_{2.5}$ and $O_3$ reductions only, in the absence of changes in baseline mortality rates and population. To clarify this, we modify the sentence in page 2 line 1-3: "We conclude that air quality improvements have significantly decreased the mortality burden, avoiding roughly 35,800 (38%) $PM_{2.5}$-related deaths and 4,600 (27%) $O_3$-related deaths in 2010 compared to the case if air quality had stayed at 1990 levels (at 2010 baseline mortality rates and population)."

2.15: observations sites –> observations
**Response:** We now deleted the "sites" in this sentence.

2.20: it's note entirely clear how the authors are separating the benefits of the NAAQS from those of the technologies put in place to meet the NAAQS – seems like these are perhaps two sides of the same coin..
**Response:** We rephrase this sentence following the reviewer's suggestions:
"Other changes in energy and emission control technology occurred concurrently with air quality regulations also helped to improve air quality."

2.24: I understand the wording, but if one accept the health impact analysis framework used here, then does it matters not where people live as much as where they die?
**Response:** Here we make the point that one needs to understand how air quality has improved in spatial relation to how population is distributed. The reviewer is correct that baseline mortality is used directly in our calculations ("where people die"), and so we have updated the language from "where people live" to "how population and baseline mortality are distributed".

2.15 - 2.24: Note the difference in tone between the assertiveness of this work ("improvements were mainly driven by ambient air quality standards…", and that of Fann 2017 who state "it is difficult to attribute this reduction to specific policy interventions….many factors are likely to have contributed….federal air quality policies are likely to have played an important role". It then seems that different federal regulations are cited, such as Acid Rain program. Thoughts about why the present work is a bit more sure of the role of regulations in this regard? Were EPA authors just being more cautious with their wording for professional reasons?
**Response:** Following the reviewer's suggestion, we now rewrote the sentence from 2.16 to 2.19:
"These air quality improvements were likely mainly driven by ambient air quality standards, and federal and state implementation of stationary and mobile source regulations, especially the 1990 Clean Air Act (CAA) Amendments, the 2002 NOx State Implementation Plans (SIP) Call, and the Cross-State Air Pollution Rule (Chestnut and Mills, 2005; U.S. EPA, 2011), together with other rules to reduce anthropogenic emissions from light duty, heavy duty, and nonroad vehicles (Fann et al.2012b; U.S. EPA2014). Other changes in energy and emission control technology occurred concurrently with air quality regulations also helped to improve air quality."

**References:**

Chestnut, L. G. and Mills, D. M.: A fresh look at the benefits and costs of the US acid rain program, J. Environ. Manage., 77(3), 252–266, doi:10.1016/j.jenvman.2005.05.014, 2005.

Fann, N., Baker, K. R. and Fulcher, C. M.: Characterizing the $PM_{2.5}$-related health benefits of emission reductions for 17 industrial, area and mobile emission sectors across the U.S., Environ. Int., 49(2012), 141–151, doi:10.1016/j.envint.2012.08.017, 2012b.

US EPA. 2014. "Control of Air Pollution from Motor Vehicles: Tier3 Motor Vehicle Emission and Fuel Standards Final Rule." https://www.epa.gov/regulations-emissions-vehicles-and-engines/final-rule-control-air-pollution-motor-vehicles-tier-3 (accessed 5 September 2018).

4.16: I doubt this is what the authors meant to say. If RRs were downloaded from the GBD web site, then there would be no role for the simulations of $O_3$ and $PM_{2.5}$ concentrations described in section 2.1.
**Response:** The relative risks (RR) for $PM_{2.5}$ from the IER function (Burnet et al., 2014) that we downloaded from the GBD website are functions of $PM_{2.5}$ concentration, and we need the gridded concentration fields to determine the RR. Also, the RRs for $O_3$ are calculated (page 4 line 17) instead of downloaded.

4.25: The justification here seems a bit odd, as if the authors decided that sticking with the mismatched population age-ranges from their previous work would take precedent over correctly matching age ranges with the epidemiological study of Jerrett 2009.
**Response:** By aligning exposed population >=25 yrs for both $PM_{2.5}$ and $O_3$, we want to determine the magnitude differences for the mortality burdens of $PM_{2.5}$ and $O_3$. This method was also adopted by Lim et a., 2012; Cohen et al., 2017. In the revised manuscript, we have rewritten the sentence as:
"We use adults above 25 years, to be comparable with other calculations of $PM_{2.5}$ mortality burdens (Lim et al., 2012; Silva et al., 2016a,b; Cohen et al., 2017), even though the estimated RR from Jerrett et al. (2009) were for adults above 30 old only."

**Reference:**
Lim, S. S., Vos, T., Flaxman, A. D., Danaei, G., Shibuya, K., Adair-Rohani, H., Amann, M., Anderson, H. R., Andrews, K. G., Aryee, M., Atkinson, C., Bacchus, L. J., Bahalim, A. N., Balakrishnan, K., Balmes, J., Barker-Collo, S., Baxter, A., Bell, M. L., Blore, J. D., Blyth, F., Bonner, C., Borges, G., Bourne, R., Boussinesq, M., Brauer, M., Brooks, P., Bruce, N. G., Brunekreef, B., Bryan-Hancock, C., Bucello, C., Buchbinder, R., Bull, F., Burnett, R. T., Byers, T. E., Calabria, B., Carapetis, J., Carnahan, E., Chafe, Z., Charlson, F., Chen, H., Chen, J. S., Cheng, A. T.-A., Child, J. C., Cohen, A., Colson, K. E., Cowie, B. C., Darby, S., Darling, S., Davis, A., Degenhardt, L., Dentener, F., Des Jarlais, D. C., Devries, K., Dherani, M., Ding, E. L., Dorsey, E. R., Driscoll, T., Edmond, K., Ali, S. E., Engell, R. E., Erwin, P. J., Fahimi, S., Falder, G., Farzadfar, F., Ferrari, A., Finucane, M. M., Flaxman, S., Fowkes, F. G. R., Freedman, G., Freeman, M. K., Gakidou, E., Ghosh, S., Giovannucci, E., Gmel, G., Graham, K., Grainger, R., Grant, B., Gunnell, D., Gutierrez, H. R., Hall, W., Hoek, H. W., Hogan, A., Hosgood, H. D., Hoy, D., Hu, H., Hubbell, B. J., Hutchings, S. J., Ibeanusi, S. E., Jacklyn, G. L., Jasrasaria, R., Jonas, J. B., Kan, H., Kanis, J. a, Kassebaum, N., Kawakami, N., Khang, Y.-H., Khatibzadeh, S., Khoo, J.-P., Kok, C., et al.: A comparative risk assessment of burden of disease and injury attributable to 67 risk

factors and risk factor clusters in 21 regions, 1990-2010: a systematic analysis for the Global Burden of Disease Study 2010., Lancet, 380(9859), 2224–60, doi:10.1016/S0140-6736(12)61766-8, 2012.

4.28 - 5.15: Would it be possible to avoid these types of inconsistencies in reporting to use heath impact functions associated with all-cause rather than cause-specific mortality rates, even if the epidemiological evidence of the responses isn't as robust?

**Response:** It is possible to avoid inconsistencies between the two standards of ICD9 and IC10 by quantifying the total all-cause mortality as used in Fann et al. (2017). However, by doing that we would assume a log-linear association between $PM_{2.5}$ and mortality burden (Krewski et al., 2009), which will have larger uncertainties when the $PM_{2.5}$ concentration is higher than 22 µg m$^{-3}$ (Burnett et al., 2014; Pope et al., 2009). From the model simulation, the $PM_{2.5}$ concentration were higher than this maximum value in 1990 in eastern US (see Figure 1a).

7.24: in the eastern

**Response:** We now add "the" in the revised manuscript.
"For other states in the eastern US"

Title: should indicate this is about ambient AQ? Or that is obvious?

**Response:** We add "ambient" before "$PM_{2.5}$" and "$O_3$" in the title as the reviewer suggested.